# Revealing DNA Structure at Liquid/Solid Interfaces by AFM-Based High-Resolution Imaging and Molecular Spectroscopy

**DOI:** 10.3390/molecules26216476

**Published:** 2021-10-27

**Authors:** Ewelina Lipiec, Kamila Sofińska, Sara Seweryn, Natalia Wilkosz, Marek Szymonski

**Affiliations:** M. Smoluchowski Institute of Physics, Jagiellonian University, Łojasiewicza 11, 30-348 Kraków, Poland; ewelina.lipiec@uj.edu.pl (E.L.); sara.seweryn@doctoral.uj.edu.pl (S.S.); natalia.szydlowska@uj.edu.pl (N.W.); ufszymon@cyf-kr.edu.pl (M.S.)

**Keywords:** high-resolution, Atomic Force Microscopy (AFM), DNA, DNA structure, molecular spectroscopy, Tip-Enhanced Raman Spectroscopy (TERS), Surface-Enhanced Raman Spectroscopy (SERS)

## Abstract

DNA covers the genetic information in all living organisms. Numerous intrinsic and extrinsic factors may influence the local structure of the DNA molecule or compromise its integrity. Detailed understanding of structural modifications of DNA resulting from interactions with other molecules and surrounding environment is of central importance for the future development of medicine and pharmacology. In this paper, we review the recent achievements in research on DNA structure at nanoscale. In particular, we focused on the molecular structure of DNA revealed by high-resolution AFM (Atomic Force Microscopy) imaging at liquid/solid interfaces. Such detailed structural studies were driven by the technical developments made in SPM (Scanning Probe Microscopy) techniques. Therefore, we describe here the working principles of AFM modes allowing high-resolution visualization of DNA structure under native (liquid) environment. While AFM provides well-resolved structure of molecules at nanoscale, it does not reveal the chemical structure and composition of studied samples. The simultaneous information combining the structural and chemical details of studied analyte allows achieve a comprehensive picture of investigated phenomenon. Therefore, we also summarize recent molecular spectroscopy studies, including Tip-Enhanced Raman Spectroscopy (TERS), on the DNA structure and its structural rearrangements.

## 1. Introduction

The DNA molecule is one of the most interesting objects to scientists due to its biological significance and properties determined by the unique chemical composition and dynamic structure. DNA, as a double helix of strands twisted around each other and held together via hydrogen bonds, was described for the first time in the early 1950s [1]. Our scientific view on DNA molecule has evolved since Crick and Watson’s historical work, as well as the Franklin and Gosling’s [2] paper published in the same issue of Nature in 1953. Franklin and Gosling’s article describes the molecular configuration of the rehydrated/hydrated sodium thymonucleate fibers based on the X-ray experimental findings, which constituted the foundation for theoretical description of helical structure of DNA. Rapid development of nanotechnology pushed the boundaries of knowledge and explored the importance of DNA local properties. Due to the local heterogeneity of DNA molecule, those analytical methods which provide information about chemical structure at the nanoscale are exceptionally beneficial.

Research on structure of biological molecules such as nucleic acids and proteins should preferably be performed under native conditions, or at least conditions mimicking their natural environments, otherwise, they may lose conformational properties, which are crucial for their biochemical activity [3,4]. However, for exploring topographical features and molecular structure at the nanoscale with all SPM techniques and their derivatives, samples need to be fixed at the flat substrate. Thus, to maintain the physiological conditions while studying the sample using SPM-based techniques, the optimal solution is deposition of studied biomolecules at the liquid—solid interface, although the interface itself could affect molecular conformation of the adsorbates. Therefore, studying biological objects at liquid-solid interfaces on one hand opens entirely new opportunities for research in a defined physiological environment, on the other hand, it poses several difficulties requiring innovative instrumental developments, as well as original data analysis methods and interpretation solutions. Furthermore, measurements undertaken in liquids open the new possibilities of monitoring structural rearrangements that occur in biomolecules upon biologically significant processes such as interactions between DNA and other macromolecules like proteins or peptides as well as chemical agents including drugs.

In this paper, we review recent achievements in revealing the molecular structure of DNA by high-resolution SPM imaging at liquid/solid interfaces. The advantages of measurements in liquid media and related technical challenges are also discussed here. The broad range of applications including studies of DNA interactions with other macromolecules and monitoring of DNA molecular dynamics are also presented. SPM provides nanometric spatial resolution; however, the molecular information that can be achieved is still limited. Therefore, this manuscript also presents a view on combination of SPM with molecular spectroscopic techniques as an important future scientific direction, which will play an important role in exploration of DNA local molecular dynamics and its role in inter and intramolecular interactions.

## 2. Preparation of DNA Samples for High-Resolution AFM Imaging

In recent years, a lot of scientific efforts have been focused on achieving high-resolution AFM images of biomolecules, including DNA molecules, DNA interacting with other small molecules or proteins. Those achievements paved the way to information that was so far elusive due to methodological limitations. High-resolution AFM imaging requires surface fixation of studied analyte. So far, various surfaces have been used for DNA imaging, for example, cationic lipid bilayers [5], gold [6,7], highly oriented pyrolytic graphite (HOPG) [8,9,10,11,12,13,14,15], and mica [16,17,18,19]. Due to the atomic flatness, mica provides favorable conditions for achieving a high contrast, which supports obtaining high-resolution AFM images. It is worth noting that except for the surface roughness, the AFM resolution is set by several factors, including the radius of a tip apex, tip-surface interactions, and environmental conditions. Mica it is the most common substrate for AFM imaging of the DNA structure [20], the interaction of DNA with small molecules [21], DNA condensation [22,23], and DNA-protein interactions [24,25]. AFM measurements in liquids enable following phenomena related to structural modifications of biomolecules in situ under physiological conditions [26]. The fixation process of negatively charged DNA on a negatively charged mica surface for high-resolution AFM imagining purposes involves the use of positively charged intermediates. DNA deposition on the mica surface is the most commonly mediated via cations or silane groups. The strength of DNA binding to the mica surface using different cations varies and can affect its structural properties, e.g., conformation of the DNA molecule. For example, it was shown by Japaridze et al. [17] in TERS studies that under the DNA fixation on mica mediated by Mg^2+^ ions, the molecule undergoes partial conformational transition at the ends of the chain upon drying. It was also found that Mn^2+^-deposited DNA did not change the length significantly with respect to its length in the bulk phase, and thus, did not rearranged the conformation. This effect was attributed to the stronger interaction between DNA and the mica surface in the case of Mn^2+^ mediated DNA deposition in comparison to Mg^2+^. The research on the ability of DNA binding to mica surface via divalent metal cations showed that the binding capacity is related to the ion radius [27,28]. On the other hand, in 1992 Hansma et al. [16] obtained the first AFM images of DNA on mica surface in liquids using magnesium ions as deposition mediators. However, in general, obtaining reproducible AFM images of DNA in liquids using Mg^2+^ ions for deposition on mica was challenging, therefore, also nickel or zinc ions were tested in terms of performing high quality AFM measurements [29]. The comparison of force-distance profiles of DNA layer deposited on mica using Mg^2+^ and Co^2+^ ions obtained with SFA (surface force apparatus) indicated that magnesium mediated DNA deposition resulted in partial adsorption with a lot of free loops or tails in the solution, while Co^2+^ enabled obtaining flat layer of DNA [30]. Normally, Ni^2+^ cations are considered to provide relatively stable and well bounded DNA layer on mica surface for high-resolution AFM imaging in liquids. Kuchuk et al. [19] reported on the gradual deterioration of the DNA adsorbed on mica using Ni^2+^ with time, therefore, the reduction of the scanning time to few minutes was needed to obtain well-resolved DNA structure. Recently, Heenan and Perkins [18] reported on the protocol of DNA fixation on mica well suited for AFM imaging in liquids without the loss of physiologically relevant information. This protocol involves the preparation of mica surface with Ni^2+^ ions, DNA deposition in buffer consisting of MgCl_2_ and KCl salts, and then imaging in the buffer containing Ni^2+^ and K^+^ ions. An application of salts at concentrations higher than physiological (0.15 M) results in aggregation of DNA molecules because of the electrostatic screening effect [31]. The second popular method of DNA binding to mica is to modify the mica surface with aminopropyl silane (APTES) [12,20,32]. The application of APTES enables both air and liquid imaging of DNA. Moreover, it was found that APTES mediated DNA deposition on mica does not affect the physiological conformation of DNA even when such a DNA layer was studied under air conditions [17]. The major drawback of using silanized mica for AFM imaging of DNA is the increase of the surface roughness in comparison to DNA samples prepared via ion deposition that may compromise obtaining high contrast, and thus high-resolution AFM images of DNA [18].

## 3. Basic Technical Developments

### 3.1. Dynamic AFM Modes

Recent progress in imaging techniques indicates that application of dynamic AFM modes with an oscillating cantilever enables reducing the pressure exerted on a studied sample by the probe apex. Therefore, such modes seem to be dedicated for measurements of soft materials, such as DNA, which are susceptible to the mechanical damage. Typically, the vertical position of the cantilever is adjusted so that the probe is periodically approaching the sample, experiencing a repulsive interaction from its surface. The interaction between the sample and the oscillating probe could cause alterations of frequency, amplitude, and phase of the oscillations. Due to sample mechanical deformations by the probe, those alternations are driven not only by the varying topography but also by mechanical properties of the investigated material. For example, based on so-called phase contrast it is possible to image not only the topography of the sample surface, but also by the physical characteristics of studied material, e.g., the stiffness heterogenicity. In general, the dynamic AFM can be realized through amplitude modulation (AM-AFM) and frequency modulation (FM-AFM). Both solutions are described in the following paragraphs. Force regimes for each AFM mode is schematically presented in Figure 1a.

#### 3.1.1. AM-AFM Imaging

In the AM-AFM mode, the probe oscillations are driven at constant frequency close to its free resonant frequency in a close vicinity of the sample surface. The amplitude of the oscillations spans the region of repulsive interaction at the tapping of the probe against the sample and the counter end of the probe trajectory experiencing an attractive interaction force. Since the contact-like interaction with the sample is only limited to small fraction of the probe trajectory, this way of operation is also known as an intermittent contact, or a tapping mode. The value of the amplitude is used as the main feedback parameter to secure the constant frequency of oscillations [33]. In Figure 1b, oscillations of the probe scanning over a DNA, modeled as a step edge, are shown schematically. Alternations of the sample topography modify the distance between the scanning tip and the sample. The free amplitude (A_r_) is reduced (to A_0_) while probe approaches the sample surface. The decrease/increase of the distance between the tip and the scanned sample, reduces (to A_1_)/increases (to A_2_) the amplitude of the cantilever oscillations. The feedback loop constantly controls the amplitude of oscillations by adjusting a piezo position in *z* direction (changing the probe-sample distance) to restore the amplitude at a predetermined value. AM-AFM measurements in liquids are technically more difficult than in air. The major limitation results from the hydrodynamic damping influence of the liquid environment on the cantilever frequency response [34]. This is due to the relatively strong interaction between the piezoelectric actuator on the entire environment of the cantilever. These excitations are reflected as multiple peaks in the cantilever frequency response plot [35].

#### 3.1.2. PeakForce Tapping

PeakForce Tapping (PFT) mode relies on force-distance curves to generate a topographic image [36]. In this mode, at each pixel (single curve) tip goes from non-contact, through attractive, and repulsive regimes to finally indent a sample. The peak force is referred to as the maximum probe–sample interaction force [37]. The individual peak force points are used to trigger the *z*-piezo retraction mechanisms. In PFT, cantilever is excited to a sinusoidal motion with a frequency, which is significantly lower than its resonant frequency. The feedback loop modulates the *z*-piezo position to keep the tip-sample interaction force constant based on the peak force from each acquired curve. The measured peak deflection (and thus force) at each point during scanning is referred to as the baseline cantilever deflection of tip being above the surface of the sample [38]. Therefore, the tip-sample interaction force is precisely controlled and the risk of the tip apex damage or sample deformation is minimized [37]. Such a precise control of the applied force enables a long-term imaging of soft biomolecules such as DNA [39].

#### 3.1.3. FM-AFM Imaging

In the frequency modulation mode (FM-AFM) [40,41], also known as a non-contact mode, the amplitude of cantilever oscillations is kept at a constant value directly by gain and phase modulation from the feedback loop. In the right panel of Figure 1b, the concept of FM-AFM is schematically illustrated. Upon measurements the cantilever vibrates at a so-called current resonant frequency (f_0_), which is different from the resonant frequency of free oscillations (f_r_) due to the sample-probe interaction. The change of the tip-sample interaction strength induces the shift of the frequency of the cantilever motions (depending on the topographical features: f_0_ → f_1_—the increase of frequency, or f_0_ → f_2_—the decrease of oscillations) [41,42]. The feedback loop, which controls the *z* position of the piezo scanner adjusts the tip-sample distance to achieve a constant value of the Δf (f_r_ − f_0_). It is important to note that the best resolution images in liquids were obtained for FM-AFM instruments operating in a repulsive force regime, so that for decreasing distance between the sample and the probe tip the frequency shift Δf is increasing [43,44].

#### 3.1.4. 3D AFM at Solid-Liquid Interfaces

The dynamic modes allow 3D data acquisition. All the above-described dynamic AFM solutions have previously been applied to image 3D solid−liquid interfaces [45]. Through the precise assessment of the probe-sample interaction force at particular distance, it is possible to provide the information not only about the properties of the sample itself, but also regarding the entire sample environment and their influence on each other.

3D AFM maps may be achieved with three different concepts of scanner movement [43,44,45]. Since, force-distance-based AFM modes provide an information about the force as a function of the tip-sample distance, the results can be presented in three dimensions (Figure 2a). A second concept of 3D imaging relies on the rendering of a stack of the 2D images acquired at selected, multiple tip-sample distances (*z* offset with fixed *xy* positions) to provide 3D visualization of the sample and its surrounding medium (Figure 2b). These two solutions to provide 3D maps are relatively slow, especially in the liquid environment, and thus, the results may suffer from the lack of sufficient resolution due to the drift resulted from possible instability of tip-sample distance in the liquid medium [43]. Since biological systems in physiological conditions are characterized by high dynamic and relatively low stability, fast and non-invasive measurements are of high importance. To improve the resolution of 3D images and time of image acquisition, the concept of scanner movement relying on the modulation of the scanner position in *z* direction with simultaneous slow lateral displacement of a piezo scanner in a fast sine wave manner is preferably applied (Figure 2c). Here, 3D AFM image is a graphical representation of oscillation amplitude/phase/frequency shift changes (depending on the applied feedback parameter) as a function of a scanner position in *x*, *y*, *z* directions [45,46]. This solution is considered to be highly relevant in the field of AFM technical developments enabling the efficient research on biomolecules.

## 4. AFM Profiling of DNA

Atomic Force Microscopy secures exceptional position among real-space imaging methods of biological structures due to its capability to operate in liquids. Recent developments of the technique, driven by the strong interest in research on biomolecules such as DNA via AFM in liquid medium since early 90s [5,16,20,47,48,49,50,51,52,53,54], clearly demonstrate its great potential not only for high-resolution imaging of DNA in aqueous solutions [18,19,55,56,57] but also for 3D observation of the hydration structures associated with individual DNA molecules deposited at the liquid-solid interface [19].

Although both AM-AFM and FM-AFM modes of operation have been used with remarkable success [18,19,55,56,57], the conventional solutions for AFM measurements are impaired by structural damage due to strong interaction forces between the tip apex and the delicate biological material, such as DNA. Despite those obstacles, recent methodological and instrumental advances described in Section 3 allowed for significant achievements in resolving geometrical and chemical structure of DNA with unprecedented accuracy. Those achievements are briefly described in the following sections.

### 4.1. Revealing Submolecular Structure of Individual DNA Molecules

Already in 2013, Ido and co-workers [56] reported on high-resolution AFM images of pUC18 DNA plasmid deposited on muscovite mica in 50 nM NiCl_2_ water solution. Using a frequency modulated mode of detection, FM-AFM, they were able to recognize major and minor grooves between phosphate groups along the sugar-phosphate backbone of B-form double helix as seen in Figure 3a. Cross sectional profiles taken along the neighboring ridges of the FM-AFM image allowed for identification of the individual phosphate groups of the DNA backbone. Local variations in molecular structure of individual regions of the plasmid were also identified. The study indicates importance of an ultra-sharp tip features with a radius estimated to be as small as 1 nm, which were occasionally formed on the probe apex facilitating exceptional resolution of the obtained topographic images.

Furthermore, Ido et al. showed that FM-AFM under liquid could resolve the submolecular structure of exemplary self-assembled DNA nanostructures [56]. This was demonstrated for the DNA tile structure constructed from commercially available single-stranded DNA (ssDNA) oligomers which in a buffer solution self-assemble into 2D hierarchical arrays of branched DNA motifs (see area marked by the white rectangle in Figure 3b). Such arrays are known in the literature as a double crossover, antiparallel junction, odd-even intramolecular spacing lattice (a DAO-E type of lattice) [58]. Among other features the authors could clearly identify the right-handedness of B-DNA used in the tile, as well as two different types of the unit connecting configurations.

High-resolution frequency shift maps of double-strand DNA acquired in FM-AFM mode were demonstrated by the group of Professor Dietler [59]. Researchers showed a double-helical groove periodicity of DNA molecule. Moreover, tip-sample van der Waals interaction at constant height between prob and sample was calculated.

Pyne et al. used both a regular AM tapping and the PeakForce Tapping modes for imaging supercoiled DNA plasmids (containing 3486 bp) [60] or DNA minicircles (339 and 251 bp) [61] supported on mica in HEPES buffer solution with NiCl_2_. Although the overall structure and dimensions of the helix could be seen right away on the acquired images, a more quantitative analysis of the data indicated their dependence on the force exerted by the probe tip. This could be well controlled in the Peak-Force Tapping mode; however, it was found that the corrugation periodicity and the helix diameter were approaching those from the B-DNA crystal structure data [62] for the lowest peak-force values only (20 nN in ref. [60]). A comparison of the images obtained in ref. [60] with the B-DNA crystal structure is reproduced in Figure 4. The selected segments of the plasmid AFM topographic image are digitally straightened and compared with the reference data. The structure of alternating major and minor grooves is clearly visible by the depth of depressions in the line profile is significantly reduced in comparison to the DNA crystal data. This indicated yet another factor influencing the AFM data, i.e., a relatively large tip probe apex diameter, limiting the resolution as previously stressed by Ido et al. [56].

Umeda and co-workers [63] used the 3D FM-AFM mapping technique in order to perform local electric double layer (EDL) force measurements on pUC18 DNA molecules in aqueous solutions of electrolyte (10 and 100 mM KCl). The molecules were deposited at mica surface coated with poly-L-lysine. It was demonstrated that the charge density measurements on DNA were feasible with molecular precision, although still in a rather specific environment.

Dynamic AM-AFM modes of imaging were also used by Ares et al. [55] for sub-helical characterization of double-stranded RNA (dsRNA) molecules adsorbed on mica substrate in a liquid environment containing Ni^2+^ ions. The helical structure of the molecule was identified as A-form and the periodic bands along the strand axis were assigned to minor and major grooves. Presumably such a double periodic structure was visible in the case of extremely sharp tip apex (a radius below 1 nm) whereas for more blunt tips, with the tip apex radius above 2.5 nm only a single periodicity could be observed. Despite the role of the tip sharpness in high-resolution imaging of dsRNA, the authors point out to minimization of the force exerted by AFM tapping probe as precondition for resolving sub-helical features of the dsRNA images.

Further advances in high-resolution imaging of individual dsDNA molecules in liquids were reported by Kuchuk and Sivan [19]. The authors used FM-AFM imaging mode in near physiological conditions identifying individual phosphate groups of the DNA backbone with very high-resolution as reproduced in Figure 5.

Heenan and Perkins [18] obtained well-resolved images of right-handed helical structure of DNA (Figure 6) in tapping mode AM-AFM under biochemically relevant conditions. High-resolution DNA imaging required an application of sharp AFM probes with 2 nm tip apex radius and scanning with a fast-scan axis being parallel to the DNA axis. The quality of AFM images enabled determining the helical pitch of DNA that was 3.51 ± 0.04 nm. It was in good agreement with helical pitch of 3.6 nm determined by Rhodes and Klug in enzyme (DNase I) digestion studies [64]. An application of physiological conditions for deposition and imaging enabled also to capture the interaction of DNA with proteins, BspMI restriction enzyme, and nucleosomes.

AM-AFM in physiological solution was applied by Tuan Phan et al. [65] in high-resolution imaging of several species of self-assembly of DNA sequence d[G4T2G4] called G-wires. These structures may find an application as conductive nanowires in nanodevices and nanoelectronics due to the long-range charge transport [66]. Their electronic properties are determined by the molecular structure, which was explored with HR AM-AFM experimental results supported by simulated AFM images and molecular dynamics, providing atomic models of G-wires that consist of several strands of d[G4T2G4]. Simulated AFM images relies on numerical deconvolution of the tip effect. This combined approach allowed several types and subtypes of G-wires to be distinguished and described. Figure 7 demonstrates the results obtained for two subtypes of type I G-wires: left-handed and zigzag [65].

Yamada et al. [57] used FM-AFM in aqueous solution for high-resolution imaging of double DNA helix in two different conformations: left-handed Z-DNA and right-handed B-DNA. Experimental results provided sizes of DNA grooves in both conformations with a high precision. Additionally, researchers analyzed the electric double layer (EDL) interactions between the probe and the sample to provide surface charge density. This was done by obtaining 3D force map through a conversion of the acquired 3D frequency shift curves (Figure 8a) to the force curves using the Sader method [67]. Then the surface charge density (Figure 8c) was calculated from a simplified model describing EDL interactions between two charged spheres: one representing probe and the second one, DNA. Figure 8 shows constant ∆f image, which corresponds to topography (Figure 8a,b) and charge density map (Figure 8c) of DNA strand with B–Z junctions, together with height and charge density profiles extracted along the strand. In this work, AFM was applied for the first time to measure the surface charge density of individual biomolecules, showing the capability of AFM in studies of more complicated systems (for example DNA-protein complexes).

### 4.2. 3D AFM Imaging of DNA Associated Hydration Structures

DNA molecules in its native environment inevitably interact with both water molecules and a plethora of other relevant atoms and molecules in various charge states [68,69,70,71,72]. The formation of layered hydration structures in a space around biomolecules, such as DNAs, determines the chemical structure of the molecules, including those adsorbed at the liquid-solid interface [69,73,74,75,76]. Furthermore, the hydration structure could be crucial for understanding the molecule functionality and details of biological processes occurring in cellular environment [19].

Over the last two decades there were numerous studies on DNA hydration; most of them, however, could not provide real-space maps of the hydration structure acquired for single DNA molecule, not to mention in the native environment. For the first time, Kuchuk and Sivan [19] reported on 3D AFM mapping of the hydration structure associated with an individual DNA molecule under liquid environment close to physiological conditions. Their findings are illustrated in Figure 9 reproduced from ref. [19]. It is seen that hydration oscillations visualized on the frequency shift of the AFM cantilever 2D cross-cuts through the acquired 3D maps are clearly resolved both on top of the DNA and over the mica substrate. Interestingly, the observed DNA hydration density was found to be mostly concentrated in areas located near the DNA grooves. Measuring the frequency shift maps could provide data for calculating the interaction force between AFM probe and DNA, i.e., a quantitative measure of the hydration strength [19]. Such information could be of high importance for understanding molecular binding mechanism to DNA, such as protein binding specific sections of the helix. The heterogenicity of hydration layer above the DNA molecule was also studied in FM-AFM mode by Santos at al. [77].

## 5. SERS and TERS Research on DNA Structure

Although SPM techniques provide highly resolved structures of biomolecules with sub-molecular resolution, they do not provide any information regarding the chemical composition. Recent developments of methodology, which combines high spatial resolution of SPM techniques with exceptionally high chemical selectivity of molecular spectroscopy, shed new light on bioresearch at the nanoscale. Now, is possible to probe local chemical structure of biological molecules. Due to the high sensitivity of Tip-Enhanced Raman Spectroscopy (TERS), this method was applied in research on DNA structure and its molecular composition.

In the following paragraph, we emphasize the achievements of spectroscopic techniques applied in liquids in structural studies of DNA. Specifically, first we discuss the limitations and difficulties related to performing spectroscopic measurements under liquid environment. Then, we describe TERS and SERS (Surface-Enhanced Raman Spectroscopy) findings in relation to DNA structure or structural modifications under the influence of external factors. While TERS relays on the combination of SPM techniques with Raman spectroscopy providing simultaneous feedback regarding the surface topography and chemical structure of studied sample, SERS belongs to the group of conventional spectroscopic techniques. SERS and TERS relay on the same phenomenon of electromagnetic field enhancement via surface plasmon resonance. The exceptional sensitivity of TERS is driven by the collective movement of electrons (surface plasmons) in the nanostructure located at the SPM probe apex. This nanostructure can be understood as an individual SERS nanoparticle. A thorough understanding of SERS principles and the acquired spectral data from biomolecules is prerequisite to apply TERS methodology in research on DNA and its complexes. Since SERS has shown a great potential in studies of DNA structure in liquids, we incorporated the findings obtained via SERS to emphasize the value of structural studies on DNA under the liquid environment. A schematic representation of TERS setup allowing an acquisition of signal from DNA is presented in Figure 10a.

Conventional spectroscopic methods (Raman or infrared spectroscopies) applied in liquid environment require significant concentration of biological analytes (about 0.5 mg mL^−1^ for DNA) to obtain sufficient SNR, which is often not physiological. Moreover, performing spectroscopic measurements in liquids remain challenging because they require acquiring the signal from: (i) a bulk phase, or (ii) a droplet. In bulk spectroscopic measurements, the acquired signal refers to the total content of functional groups in molecules, thus the local heterogeneity of studied sample cannot be detected. On the other hand, collecting spectroscopic signal from droplets requires the use of relatively low laser power (usually below 10 mW, depending on the instrumentation) to prevent the droplet from drying and/or the increase of the analyte concentration related to evaporation of liquid because of the increase of temperature associated with applied laser power. The evaporation of liquid phase leads to the increase of buffer/salt concentration, and in consequence, the change of e.g., ionic strength or pH. The ionic strength is crucial for inter- and extra-molecular interactions, it defines the range of electrostatic interactions (Debye length). Thus, the change of ionic strength related to water evaporation may affect the inter-molecular structure of the studied analyte. Thus, in such a case, the observed structural modification could be easily misinterpreted as the expected alternations related to the application of e.g., chemical factors influencing DNA structure. These aspects are crucial in structural studies of DNA, which adopt specific conformations depending on the environmental properties, for example B-DNA form is stable only at physiological concentration of salt and neutral pH [78]. On the other hand, an application of relatively low laser power may result in insufficient SNR. Too high laser power can be destructive [79], especially for biological samples such as DNA.

Infrared spectroscopy research on DNA is mainly carried out in the mid infrared range because it covers vibrations of DNA marker bands from DNA backbone (phosphate motions), and vibrations of purine/pyrimidine bases. However, it poses challenging to obtain biologically relevant results while studying DNA in liquid environment because of the high absorption of the infrared light by water molecules. Therefore, Raman spectroscopy seems to be an efficient technique in the structural research on the DNA molecule in liquids. However, for analytes studied in physiological range of concentrations, conventional vibrational spectroscopy displays insufficient sensitivity. In Raman spectroscopy, only 1 per 10^10^ photons is scattered inelastically [80,81]. Moreover, biological molecules such as DNA or proteins show relatively low cross-section for Raman scattering. Therefore, to follow structural modifications in biological molecules under native conditions it is reasonable to apply plasmonic techniques such as SERS or TERS, which allow enhancing the naturally weak signal from biomolecules and to perform measurements in liquid environment. In the present paragraph, we summarized SERS and TERS achievements in the research on DNA structure.

Plasmonic techniques such as SERS were already shown to display a great potential to probe structure of the DNA molecule under native, liquid conditions. SERS in liquid environment was incorporated by Panikkanvalappil et al. [82] to follow structural modifications of DNA inducted by reactive oxygen species (ROS). It is extremely important to perform all DNA-related structural studies under liquids to prevent from dehydration-related conformational transitions in DNA structure. SERS spectra recorded for DNA in the presence of Ag nanoparticles after the exposure to H_2_O_2_/UV light revealed structural modifications accompanying double-strand breaks (DSBs) inducted via ROS attack in DNA molecule. The major structural alternations related to DNA damage were observed for bands assigned to DNA backbone and base unstacking. The band at 1085 cm^−1^ specific for the geometry of phosphate backbone shifted to 1075 cm^−1^. Moreover, a significant increase of the intensity of this band was observed. These spectral differences confirmed the conformational rearrangements of DNA backbone under the H_2_O_2_/UV exposure. The second spectral marker of ROS-inducted DNA damage was a split of the band at 740 cm^−1^ into two bands at 715 cm^−1^ and 738 cm^−1^, which was attributed to the presence of unpaired adenine and thymine residues. The described spectral differences have been explained by authors as a consequence of the hydrogen atom abstraction by the ROS.

Yue et al. [83] also studied via SERS in a liquid environment the effect of ROS on DNA damage. The ROS-inducted cell damage was observed at the level of cellular organelle, precisely mitochondria isolated from phototherapy-treated cells. For amplification of Raman signal, gold nanorods (AuNRs) were used. The most prominent spectral change related to the damage of DNA backbone was the intensity decrease and a shift toward lower energies of the band at 1092 cm^−1^ attributed to phosphate symmetric stretching.

Barhoumi et al. [84] studied spectral pattern of DNA after the exposure to cisplatin, an anticancer drug known to affect DNA replication [85], transcription [86,87,88,89], and to trigger cell apoptosis [86]. SERS spectra were acquired in full immersion of sample and objective using 63× water immersion lens and Au nanoshell-based substrates to enhance Raman signal. The incubation of DNA with cisplatin resulted in a band at 450 cm^−1^ at SERS spectra attributed to the platinum-amine stretching mode revealing the covalent bonding between cisplatin and DNA.

Further research considering interaction of cisplatin with the DNA molecule was conducted by Masetti et al. [90]. Positively charged spermine-coated silver nanoparticles (AgNp@Sp) were used as SERS enhancer. Based on spectral data, it was found that cisplatin forms covalent bond with N7 atom from guanine.

Several studies concerning interaction of another chemotherapeutic drug—doxorubicin with the DNA molecule have been performed as well. Morjani et al. [91] used SERS for selective measurements of doxorubicin in complex with Calf thymus DNA and in living cancer cell. Silver hydrosol was applied as the Raman signal amplifier, precisely silver nanoparticles were used as a SERS substrate. The authors demonstrated that SERS spectra obtained from in vitro doxorubicin-DNA complex correlate with the data acquired from nucleus of living cells treated with the anticancer drug. Beljebbar et al. [92] exploited aqueous silver hydrosol in the direct SERS analyses of doxorubicin-DNA complex. The decrease in intensity of the bands at 1214 and 1246 cm^−1^ assigned to in-plane C–O, C–O–H vibrations and C–H bending confirms the intercalation of doxorubicin rings within the DNA structure. The studies on DNA-doxorubicin binding process using synthesized nano-scale particles with probe/target oligonucleotides as a SERS substrate were continued by Spadavecchia et al. [93]. Presence of the doxorubicin-DNA complex was confirmed by appearance of the band at 1591 cm^−1^ in the SERS spectrum. Based on the decrease in the band intensity at 1586 cm^−1^ the authors suggested that the doxorubicin interaction sites were C–O–NH and ring phenyl group. Recently, Kang et al. [94] used plasmonic-tunable Raman/fluorescence imaging spectroscopy (P-TRFIS) for tracking the release and delivery of doxorubicin to the lysosomes at the single living cell level. Gold nanoparticles were applied as the enchantment factor for SERS measurements and drug carriers. Tuning the Raman and fluorescence signals of doxorubicin molecules by plasmonic field of gold nanoparticles enabled selective switching “ON” and “OFF” the Raman and fluorescence signals of doxorubicin. This approach can be a useful tool for tracing molecular mechanisms of drug delivery.

SERS in liquid environment was also applied in biomedical applications. Lin et al. [95] for the first time used SERS-based nanotechnology for the so-called liquid biopsy to detect nasopharyngeal cancer (NPC). Negatively charged silver nanoparticles were used as enhancer for DNA-SERS measurements. Quantitative SERS detection of nucleobases combined with principal component analysis (PCA) and linear discriminant analysis (LDA) was applied for the detection of blood circulating DNA. Nasopharyngeal cancer was detected with diagnostic sensitivity of 83.3% and specificity of 82.5%. Zeng et al. [96] used the new readout technique—“Click” SERS, for detection of the DNA molecule. Enhancement of the Raman signal was derived using gold nanoparticles assemblies. This innovative technique relies on identification of multiple biomarkers under a single scan and brings new insights into cancer diagnosis and treatment.

SPM-based Raman nanospectroscopy, specifically Tip-Enhanced Raman Spectroscopy (TERS), has never been applied in the research on DNA structure under liquid environment so far. The methodology exploiting the enhancement of Raman signal on plasmonic nanostructure on SPM probe provides a sensitive, non-invasive, and label-free tool to obtain chemical information with nanometer-scale spatial resolution [97,98]. TERS has already been shown to probe the DNA structure with nanometric spatial resolution, but all these studies have been carried out in air mainly due to methodological limitations. For example, TERS studies revealed that DNA double-strand breaks (DSBs) resulting from ultraviolet-C (UV-C) treatment are related to the breakage of particular bonds (Figure 10b) [99]. The TER spectra corresponding to rupture of the bands are shown in in Figure 10c. From the DNA lesions sites three types of spectra indicating different molecular modifications were acquired. Based on the abundance of the spectra (Type 3) it was proved that C–O bond is the most susceptible to UVC induced damage [99]. This SPM-based plasmonic technique enabled also to obtain spectral signature of single- or double-stranded DNA showing that TERS is efficient to probe at nanoscale organized DNA-based nanostructures [100]. Lots of scientific efforts have been recently focused on proving that TERS could be applied to reveal the sequence of nucleobases in nucleic acids, DNA and RNA [101,102,103,104,105]. TERS studies enabled distinguishing the signal obtained for plasmid-embedded DNA and plasmid-free counterparts because of different surface density of nucleobases in these two cases, which was higher in the presence of plasmid [97]. In this work, plasmid-embedded DNA was the circular plasmid molecule with its sequence of nucleobases an insert of linear DNA fragment of known sequence (β2AR). Plasmid-free DNA was the liner β2AR DNA fragment obtained by enzyme digestion (with HindIII and XbaI) of plasmid molecule. Moreover, TERS has been shown to have high capacity in probing the effect of DNA-surface interactions [17,106]. TERS measurements revealed that DNA deposited on mica mediated via divalent cations (Mg^2+^ or Mn^2+^) undergoes local conformational transition from B- to A-form at the ends of DNA chain under drying, while the middle part of DNA molecule retained B-DNA conformation [17]. TER signature of DNA deposited on silanized mica confirmed that surface silanization preserves B-DNA conformation, but due to the strong interactions of DNA with silane groups, markers of APTES–DNA interaction were affecting DNA signature [106].

Nevertheless, all these achievements in studies of DNA structure at nanoscale via TERS, even though carried out in the air, point out the important direction in the research on structural modifications in the DNA molecule. Future developments in this field should be focused on searching methodological and technical solutions that allow to perform analogical experiments but under liquid conditions. The TERS findings obtained in liquid for amyloid-β showing the rearmament of secondary structure from protofibrils to fibrils conversion in the aggregation process, already proved the great potential of this technique applied in liquid environment in the research on structural transitions of biomolecules [107].

## 6. Protective Role of Solvents

Measurements in liquids are especially beneficial for nanospectroscopic experiments. Solvents prevent or largely prohibit damage and decomposition of samples, which may occur in strong, local electromagnetic field generated in TERS and SERS. The quality of spectral data precisely SNR is better for data acquired in liquid than in ambient conditions. The protective role of liquids is complex and takes advantage from several temperature dependent physical phenomena and physical properties of solvents, plasmonic nanostructures, samples, and substrates [107].

In both SERS and TERS, excitations of surface plasmon resonance (SPR) generate the local increase of temperature. According to theoretical predictions and experimental results the temperature increase in the plasmonic hotspots depends on the amounts of absorbed and dissipated energy, and it is in the range from several dozens to hundreds degrees [108,109]. The high heat capacity and thermal conductivity of liquids causes efficient heat dissipation from the plasmonic hotspots, resulting in a decrease of the effective temperature of the plasmonic nanostructures: TERS probes or SERS active nanoparticles.

The local heating may induce sample thermal decomposition through pyrolysis or molecular desorption [109,110]. Delicate biological samples such as nucleic acids are particularly vulnerable. These processes support the formation of carbonaceous species and their accumulation on TERS probes and SERS substrates. The accumulation of carbonaceous contaminations on plasmonic nanostructures is a dynamic process, which involves a formation of aromatic rings and their transformations via rapid C-C or C-O bond creation [98]. Therefore, the spectral signal from this carbon network is observable as rapidly fluctuating peaks, which averaged over many spectra, resulting in a broad D-band (1360 cm^−1^) and G-band (1580 cm^−1^) [98]. Sample thermal decomposition and accumulation of amorphous carbon can be effectively reduced by heat dissipation from the plasmonic hotspots.

The local heating drives the thermal diffusion of surface atoms in and out of the plasmonic nanostructures. Surface atoms can absorb and desorb, destabilizing the crystal facets of the metal nanostructures and affecting the stability of the TER or SER signals [111].

In plasmonic hotspots hot carriers are generated due to nonradiative decay of surface plasmons via Landau damping and also thermal excitations [112,113]. Sample damage might be caused by the transfer of plasmon-induced hot carriers from the plasmonic nanostructures to the analyzed molecules. Hot electrons enable the dissociation of chemical bonds in small diatomic molecules as well as complex organic macromolecules [114,115,116]. The temperature determines the reaction rate of the plasmon-induced dissociation because of a coupling between electronic and vibronic excitations [114,115,116]. Therefore, the desired decrease of the probe temperature reduces this process or slows it down.

Hot electrons can be transferred from the surface of plasmonic nanostructures to the unoccupied orbitals of sample or medium atoms, resulting in a generation of reactive anions or radicals [117,118]. Hot holes may also contribute to the formation of reactive oxygen species, for example via oxidation of OH^−^ ions or H_2_O molecules, which lead to the generation of hydroxyl radicals [118]. The oxygen and hydroxyl radicals can also be responsible for the degradation of the analytes [118]. The availability of such radicals in ambient is higher than in liquids due to their different lifetimes and diffusion coefficients that determine a transport distance (diffusion length). Therefore, the molecules of the samples immersed in water or physiological buffers are less exposed to the destructive influence of radical species than the molecules in ambient air [118].

## 7. Concluding Remarks and Future Outlook

In this review, we presented recent achievements in liquid HR-AFM imaging of DNA, demonstrating applicability of this methodology in deep investigation of not only DNA topographical features but also its properties and the local molecular structure. Presently, it is possible to provide periodicity of helical structure revealing DNA conformations. Applications of recently developed 3D AFM at solid-liquid interfaces allowed the local surface charge density and local hydration of DNA to be calculated. The importance of samples preparation is highlighted here and various methods of DNA fixation on flat surfaces are described. In addition, we briefly summarized the technical aspects and working principles essential to understand high capability of the liquid scanning probe microscopy in studies of nucleic acids and their complexes with other biomolecules.

Scanning probe microscopy provides extremely high spatial resolution and delivers unique information about DNA topographical features and its properties but the structural information is still limited. Broad perspectives will involve a combination of HR-SPM with other experimental methods that deliver direct information about chemical structure and composition. One of significant future directions is simultaneous analysis of topographical features and their molecular vibrations via AFM or STM combined with chemically selective molecular spectroscopy such as infrared or Raman. This methodology is universal, and it will be applied in studies of numerous biological systems such as DNA, DNA-proteins or, DNA-drugs complexes, chromatin, multi-component lipid membranes and many others.

This review summarizes advantages of measurements in liquid and physical phenomena responsible for the beneficial role of solvents in molecular nanospectroscopic techniques, highlighting a very important future direction in DNA investigations at liquid/solid interface. Measurements in conditions that mimic nucleic acids natural environment will allow following in real time biologically significant processes that are of key importance for life. New findings about the dynamics of structural changes in DNA molecule upon damage, repair, and interaction with other biomolecules are fundamentally important in understanding the mysterious role of DNA conformational transitions and chromatin integrity in formation of DNA damage, DNA repair pathways, and the ability to participate in biochemical reaction crucial for life.

## Figures and Tables

**Figure 1 molecules-26-06476-f001:**
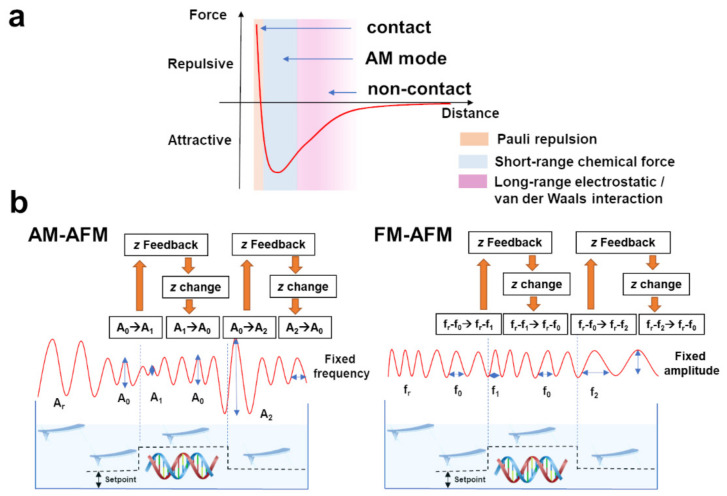
(**a**) Force regimes for contact, AM-AFM and FM-AFM. (**b**) Schematic representation of the working principles of AM-AFM (left panel) and FM-AFM (right panel) modes.

**Figure 2 molecules-26-06476-f002:**
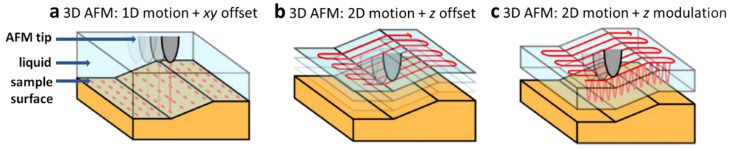
The concepts of scanner movement in 3D AFM imaging: (**a**) scanner is moving in a 1D motion with *xy* offset, (**b**) scanner moves at a *xy* plane and a given *z* offset, (**c**) scanner moves at a fast sinusoidal motion in *xy* direction with simultaneous modulation of *z* position. Reproduced with permission from Fukuma, T.; *ACS Nano*; published by ACS Publications, 2018.

**Figure 3 molecules-26-06476-f003:**
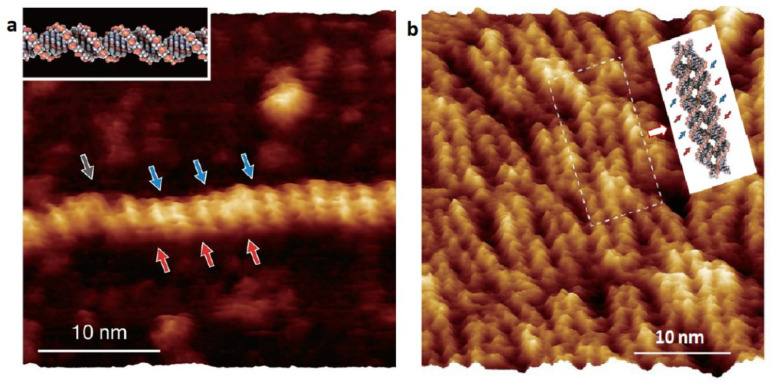
High-resolution FM-AFM topographic images of (**a**) the B-DNA plasmid and (**b**) the 2D DNA tile structure in aqueous solution of 50 nM NiCl_2_. Well-resolved positions of major and minor grooves are marked with red and blue arrows respectively. (**a**) The damaged intersection is marked by the gray arrows. A model molecular structure of the B-DNA is shown in the inset. (**b**) A theoretical model of the DNA tile unit with four branches (two at top of the unit and two at the bottom) corresponding to the image area indicated by the white rectangle is shown in the upper right corner of the panel. Composed from original figures reproduced with permission from Ido, S.; *ACS Nano*; published by ACS Publications, 2013.

**Figure 4 molecules-26-06476-f004:**
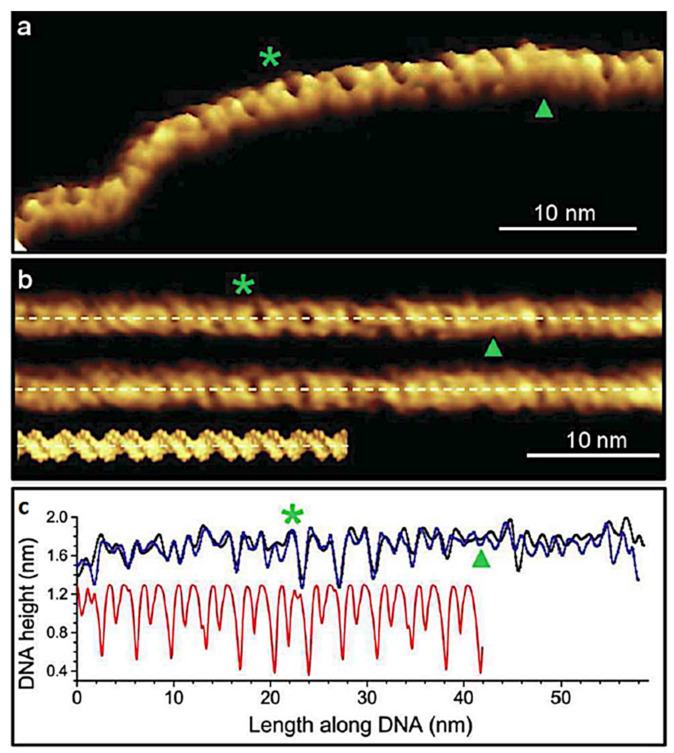
The analysis of Peak-Force Tapping mode AFM image of the B-DNA plasmid segment. The positions of the major and minor grooves are marked by green asterisks and triangles respectively. The color scale representing the height is saturated and extending up to at 1.1 nm. The digitally straightened trace (upper image) and retrace images of the segment taken from the topographic image in (**a**) are compared with a space-filling representation of the B-DNA crystal structure in (**b**). The respective height profiles along the dashed lines indicated in (**b**) are shown in (**c**). Images reproduced under the CC BY license from Pyne, A.; *Small*; published by Wiley-VCH GmbH, Weinheim, 2014.

**Figure 5 molecules-26-06476-f005:**
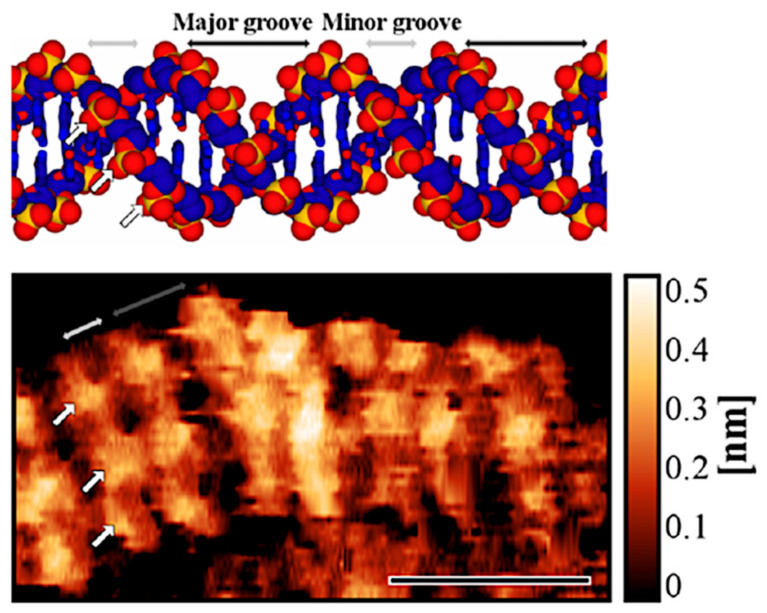
(**Upper panel**) A model of DNA in a B-conformation; (**Lower panel**) Ultrahigh-resolution AFM image of the DNA molecule obtained in FM-AFM mode. White and gray arrows on DNA model and topography are indicating the major grove, minor groove, and phosphate groups of the DNA backbone. Scale bar: 5 nm. The images are reproduced with permission from Kuchuk, K.; *Nano Lett.*; published by ACS Publications, 2018.

**Figure 6 molecules-26-06476-f006:**
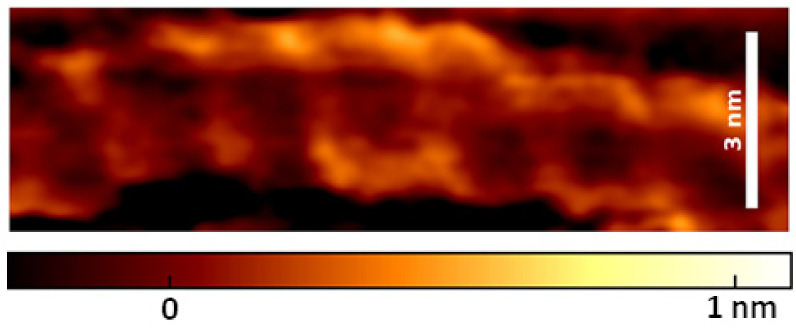
High-resolution AFM image of DNA obtained in tapping mode using a tip with 2 nm of apex radius. Reproduced with permission from Heenan, P.R.; *ACS Nano*; published by ACS Publications, 2019.

**Figure 7 molecules-26-06476-f007:**
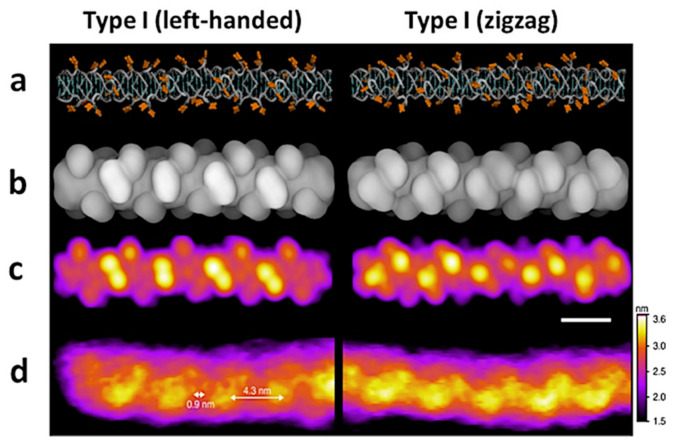
An experimental and theoretical results of molecular structure visualization of type I G-wires showing two types of periodic features: left-handed and zigzag: (**a**) the structural model, thymines are marked in orange. (**b**) Surface representation of (**c**) a simulated AFM images, (**d**) HR AM-AFM, the length of scale bar is 4 nm. Images reproduced under the CC BY license from Bose, K.; *Nat. Commun.*; published by Springer Nature, 2018.

**Figure 8 molecules-26-06476-f008:**
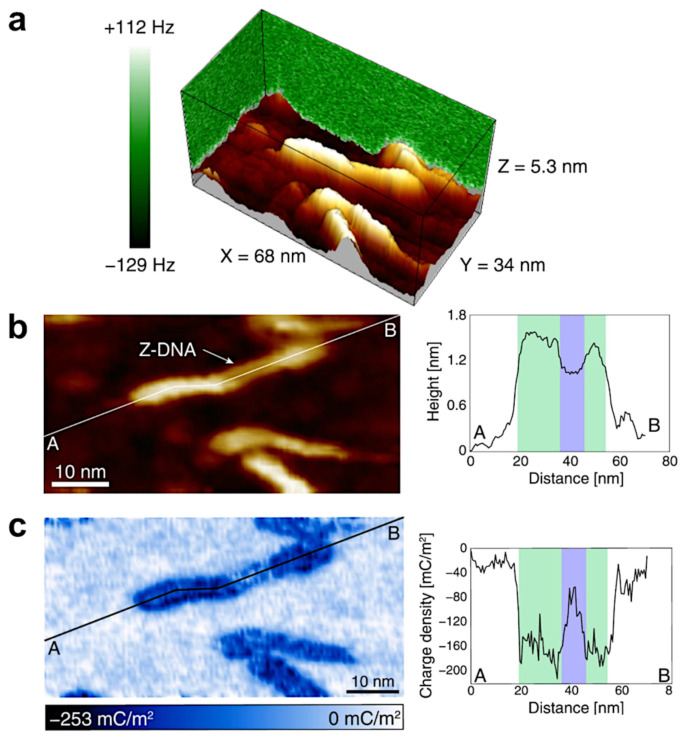
(**a**) 3D frequency shift map; (**b**) constant ∆f image corresponding to topography image, (**c**) charge density map of DNA strand with B–Z junctions. The height and charge density profiles are extracted along the lines marked on corresponding maps. Images reproduced under the CC BY license from Kominami, H., *Sci. Rep.*; published by Springer Nature, 2019.

**Figure 9 molecules-26-06476-f009:**
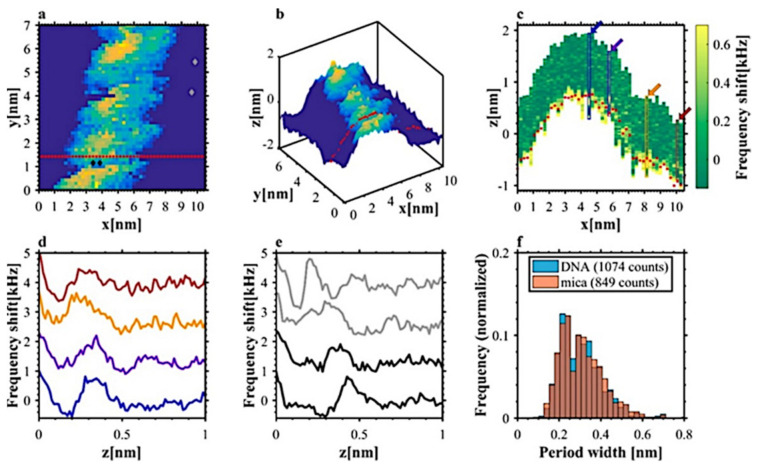
3D FM-AFM image of the DNA section measured as a set of 50 × 50 frequency shift vs. distance curves. 2D constant frequency shift (300 Hz) representation of the image is presented in (**a**). The color scale reflects the z-height as seen on 3D view in (**b**) panel. The frequency shift cross-section along the red-dot line in (**a**,**b**) are shown in panel (**c**). The individual frequency shift vs. distance curves taken along the matching color vertical rectangles indicated in (**c**) are plotted in panel (**d**). The curves are shifted vertically to facilitate the comparison. The comparison between frequency shift curves over DNA (marked in black on (**e**) and black diamonds on (**a**)) and the curves over mica substrate (marked in gray on (**e**) and gray diamonds on (**a**)) clearly show the hydration oscillations with the periods compared on histogram (**f**). Reproduced with permission from Kuchuk, K., *Nano Lett.*; published by ACS Publications, 2018.

**Figure 10 molecules-26-06476-f010:**
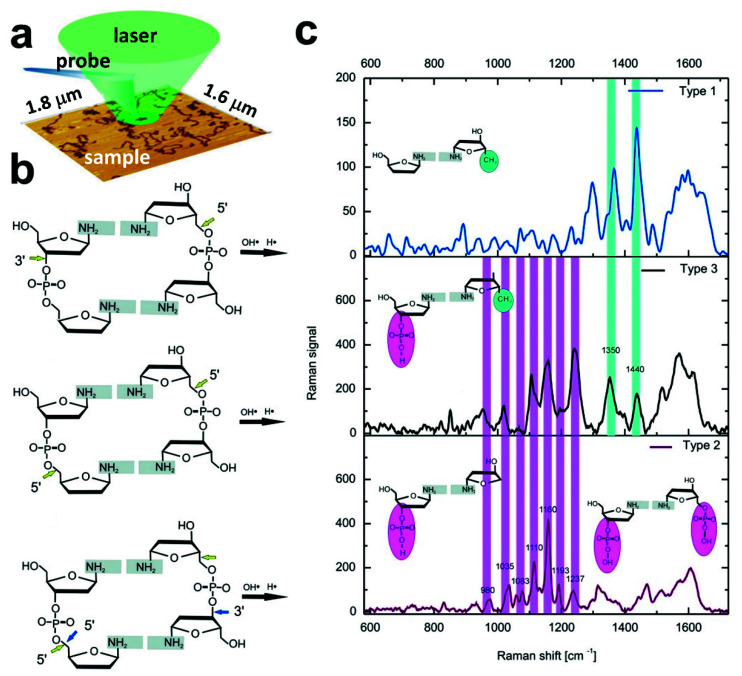
(**a**) A schematic representation of the TERS up-right setup suitable for DNA measurements. (**b**) Possible mechanisms of DNA double-strand breaks formation. (**c**) Spectra acquired from DNA lesions sites indicating DNA backbone rupture according to schemes presented in (**b**). Reproduced with permission from Lipiec, E.; *Angew. Chem.—Int. Ed.*, published by Wiley-VCH GmbH, Weinheim, 2014.

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
