# Peer review of "Revealing DNA Structure at Liquid/Solid Interfaces by AFM-Based High-Resolution Imaging and Molecular Spectroscopy"

_molecules, 2021, doi:10.3390/molecules26216476_

Round 1

Reviewer 1 Report

The manuscript by Lipiec at al. summarizes in great detail studies focused on understanding the molecular organization of DNA on surfaces. More specifically, the authors focus on two techniques, atomic force microscopy (AFM) and scanning probe microscopy (SPM) used to unravel the sub-molecular structure of DNA. The authors describe in great detail how the various modes of AFM technique work and what are the challenges associated with each of them. Then the authors move on to discussing recent spectroscopy studies, including Tip-enhanced Raman Spectroscopy (TERS) and its use in understanding DNA structure. Final part discusses the effect of solvent in achieving high spectral signal to noise ratio while preserving the DNA structure.

The review covers recent literature on the topic well and highlights technical advancements in the field. It is reasonably well written and while there are several review papers already in literature describing how the AFM and SPM  techniques work, this review focuses on sub-molecular DNA organization which is quite new and insightful.

I can recommend the paper for publication, however, strongly encourage the authors to improve the remaining points, see below.                              

1. There is no clear connection between the AFM and SERS-TERS parts. AFM focuses on DNA sub-molecular structure, while SPM part focuses also on the interaction of DNA with external factors. The focus also comes heavily on the AFM part, as evidenced by 9 images out of 10 are AFM data. Can the authors expand a bit on the ters aprt and have the flow of story more connected?

2. While some figures are very clear and informative to the reader, some need improvement. Improving them would help inexperienced reader understand the context easier and would greatly increase the quality of the manuscript.

The figure 1: It would benefit from showing a force distance curve to explain to the reader what kind of forces the AFM tip feels before going into details of amplitude/frequency modulation.  The text in the boxes are also quite hard to read.

Figure 2: The text is too small and pixelated. Neither sample, surface, tip, nor liquid above are indicated. The complicated geometry of the sample surface makes it unnecessarily complicated for the reader to follow the scanning modes.

Figure 3: I would remove the right panel as it is confusing. It is unclear where the height profile along the DNA is measured. It would be intuitive to show a line rather than a rectangle. Also it is unclear why the height of the DNA is measured in a.u and not in nm ?

3. I recommend expanding the future outlook part at the end of the manuscript. It is too short and focused on summarizing the manuscript rather than giving a bigger and more general perspective on the field.

Minor Points:

Add references for Figures 1, 4, 7, 8 in captions.

References:

Some of the important references are missing and I recommend adding them :

When discussing the 1953 paper from Crick and Watson [1], one should also mention the experimental work by Franklin and Gosling [in the same issue of Nature in 1953] as it laid the foundation for their theoretical prediction.

When discussing imaging DNA sub-molecular structure by AFM, I would recommend citing Cerreta et al (2013) European polymer journal 49.8, 1916-1922.

When discussing the importance of hydration layer when imaging DNA by AFM, one should cite: Santos et al. (2012) The Journal of Physical Chemistry C 116.4. 2807-2818.

I would recommend citing a recent paper by Pyne et al. (2021) Nature communications 12.1. 1-12.

Pg. 2 line 71: I would recommend also citing earlier works by the Dietler lab (EPFL) on studying DNA deposition on HOPG by AFM.

Reference 12 & 22 are the same.

Other Minor points:

SNR is defined multiple times in the manuscript.

Pg. 1 line 11: It says : ‘modifications of DNA resulted from interactions’, should be ‘resulting from’.

Pg. 1 line 16: It says: ‘SPM’, without defining it.

Pg. 1 line 30: It says: ‘double helix of two strands’, the word two is unnecessary.

Pg. 2 line 65: AFM is defined twice, in the abstract as well as the main text.

Pg. 2 line 71-73: The authors write: ‘Due to the atomic flatness, mica provides favorable conditions for achieving a high signal-to-noise ratio (SNR), and therefore, high-resolution AFM imaging.’. It is unclear what the authors mean by signal to noise ratio. The AFM resolution is set by several parameters, including the surface roughness, but also the tip radius, tip-surface interaction and environmental condition. I would recommend adding a sentence discussing the limits of AFM resolution.

Pg. 2 : When discussing the influence of ions on DNA deposition , I would recommend also citing: Pastré et al. (2006). Langmuir 22, 6651–6660.

Pg. 3 line 148 : The authors write : ‘The major limitation results from the influence of the liquid environment on the cantilever frequency’, do they mean cantilever damping or something else?

Pg. 4 line 147 : it is unclear what f2 relates to.

Pg. 6 line 251: I would recommend adding the original images of the DAO-E type lattice, as it it hard to follow the discussion without reading the original paper.

Pg. 8 line 307: The authors mention the helical pitch of the DNA 3.51 ± 0.04 nm, without mentioning the expected pitch for the B-form DNA.

Pg. 9: Figure 6: panels b & c show DNA –protein complexes at low resolution therefore I would suggest removing them. They are not in line about the discussion of resolving the sub-molecular structure of the DNA.

Pg. 10 line 340: It is unclear what the bottom and upper panels refer to on figure 8, as there are four of them.

Pg. 10 line 341-44: In this work, AFM was applied for the first time to measure the surface charge density of individual biomolecules, highlighting the capability of AFM in such studies of more complicated systems such as for example DNA-protein complexes.

Pg. 12 line 401: The authors mention significant concentration , without giving a rough concentration. Would be nice if it is added.

Pg 13 line 407: low laser power. Can the authors give an typical range?

Pg 13 line 430: ‘1 per 1010 photons’, I think there is a typeo, since this number is very low.

Pg 13 line 442: ‘Authors of this work took special attention to prevent …’ it is unclear what is mean by special attention. Can the authors clarify it?

Pg 14 line 462: When discussing the mode of action of cisplatin, one should also mention that it also interferes with transcription. One should also mention the relevant citation.

Pg 14 line 463: The authors mention the magnification of the lens (63x) used in the study. IS it somehow relevant?

Pg 14 line 474: It is briefly mentioned that silver hydrosol was used as signal amplifier. How does it work? And is it commonly used for Raman measurements?

Pg 14 line 491: F missing in ON and OFF.

Pg 15 lines 521-23: It is unclear what plasmid-embedded DNA and plasmid-free DNA mean and what is meant by nucleobase density. The authors should clarify it in the text.

Pg 15 figure 10- X and Y axis on the figure are barely visible. Tip, sample, laser are not indicated on the schematics.

Pg 16 Chapter 6: Is very well written and contains basic description of the techniques, therefore I would recommend placing it before chapter 5. Of course it is optional.

Pg 16 line 551: ‘’ in liquids that’, should be ‘ in liquid than

Author Response

Re: Submission ID molecules-1418119

Revealing DNA Structure at Liquid/Solid Interfaces by AFM-Based High-Resolution Imaging and Molecular Spectroscopy

Ewelina Lipiec, Kamila Sofińska*, Sara Seweryn, Natalia Wilkosz and Marek Szymonski

Reply to Reviewer 1 comments:

The manuscript by Lipiec at al. summarizes in great detail studies focused on understanding the molecular organization of DNA on surfaces. More specifically, the authors focus on two techniques, atomic force microscopy (AFM) and scanning probe microscopy (SPM) used to unravel the sub-molecular structure of DNA. The authors describe in great detail how the various modes of AFM technique work and what are the challenges associated with each of them. Then the authors move on to discussing recent spectroscopy studies, including Tip-enhanced Raman Spectroscopy (TERS) and its use in understanding DNA structure. Final part discusses the effect of solvent in achieving high spectral signal to noise ratio while preserving the DNA structure.

The review covers recent literature on the topic well and highlights technical advancements in the field. It is reasonably well written and while there are several review papers already in literature describing how the AFM and SPM  techniques work, this review focuses on sub-molecular DNA organization which is quite new and insightful.

I can recommend the paper for publication, however, strongly encourage the authors to improve the remaining points, see below.

  1. There is no clear connection between the AFM and SERS-TERS parts. AFM focuses on DNA sub-molecular structure, while SPM part focuses also on the interaction of DNA with external factors. The focus also comes heavily on the AFM part, as evidenced by 9 images out of 10 are AFM data. Can the authors expand a bit on the ters aprt and have the flow of story more connected?

We expanded the first two paragraphs of section “5. SERS and TERS research on DNA structure” to make the text flow more consistent and to give readers a clear connection between AFM, SERS and TERS sections. For details, please see pg. 13-14 lines 414-435.

  1. While some figures are very clear and informative to the reader, some need improvement. Improving them would help inexperienced reader understand the context easier and would greatly increase the quality of the manuscript.

The figure 1: It would benefit from showing a force distance curve to explain to the reader what kind of forces the AFM tip feels before going into details of amplitude/frequency modulation. The text in the boxes are also quite hard to read.

Figure 1 is corrected, text in boxes is larger now and we added force-distance curve demonstrating force regimes in AFM modes.

Figure 2: The text is too small and pixelated. Neither sample, surface, tip, nor liquid above are indicated. The complicated geometry of the sample surface makes it unnecessarily complicated for the reader to follow the scanning modes.

Figure 2 is corrected. The quality of text is improved, moreover, sample surface, liquid and AFM tip are indicated now.

Figure 3: I would remove the right panel as it is confusing. It is unclear where the height profile along the DNA is measured. It would be intuitive to show a line rather than a rectangle. Also it is unclear why the height of the DNA is measured in a.u and not in nm ?

We removed the right panel from Figure 3. Indeed, it is not clear why authors of the cited paper gave the y axis in a.u. It is not clarified within the text. All other cross-section profiles in this article are given in nm.

  1. I recommend expanding the future outlook part at the end of the manuscript. It is too short and focused on summarizing the manuscript rather than giving a bigger and more general perspective on the field.

Corrected. We expanded future outlook part. Please see pg. 19-20, lines 652-671.

Minor Points:

Add references for Figures 1, 4, 7, 8 in captions.

Figure 1 is the original figure prepared by us especially for this review. References for Figures 4, 7, and 8 are added.

References:

Some of the important references are missing and I recommend adding them :

When discussing the 1953 paper from Crick and Watson [1], one should also mention the experimental work by Franklin and Gosling [in the same issue of Nature in 1953] as it laid the foundation for their theoretical prediction.

As suggested, we added the work of Franklin and Gosling with a short description, please see pg. 1, lines 31-37.

When discussing imaging DNA sub-molecular structure by AFM, I would recommend citing Cerreta et al (2013) European polymer journal 49.8, 1916-1922.

As suggested, we cited this paper.

When discussing the importance of hydration layer when imaging DNA by AFM, one should cite: Santos et al. (2012) The Journal of Physical Chemistry C 116.4. 2807-2818.

We added this paper.

I would recommend citing a recent paper by Pyne et al. (2021) Nature communications 12.1. 1-12.

We added this paper.

Pg. 2 line 71: I would recommend also citing earlier works by the Dietler lab (EPFL) on studying DNA deposition on HOPG by AFM.

We added four references of Prof. Dietler on studying DNA deposited on HOPG via AFM. Please see pg. 2 line 77.

Reference 12 & 22 are the same.

Corrected. Now it is one ref. 17.

Other Minor points:

SNR is defined multiple times in the manuscript.

Corrected. Now SNR is defined once, when it is mentioned for the first time.

Pg. 1 line 11: It says : ‘modifications of DNA resulted from interactions’, should be ‘resulting from’.

Corrected.

Pg. 1 line 16: It says: ‘SPM’, without defining it.

Corrected. Now ‘SPM’ is defined in the abstract.

Pg. 1 line 30: It says: ‘double helix of two strands’, the word two is unnecessary.

Corrected. The word ‘two’ is removed.

Pg. 2 line 65: AFM is defined twice, in the abstract as well as the main text.

Corrected, now AFM is defined only in the abstract.

Pg. 2 line 71-73: The authors write: ‘Due to the atomic flatness, mica provides favorable conditions for achieving a high signal-to-noise ratio (SNR), and therefore, high-resolution AFM imaging.’ It is unclear what the authors mean by signal to noise ratio. The AFM resolution is set by several parameters, including the surface roughness, but also the tip radius, tip-surface interaction and environmental condition. I would recommend adding a sentence discussing the limits of AFM resolution.

In the previous version of the manuscript, we used SNR in relation to a contrast of SPM images. According to the Reviewer comment, we corrected the text: “Due to the atomic flatness, mica provides favorable conditions for achieving a high contrast, which supports obtaining high-resolution AFM images. It is worth noting, that except the surface roughness, the AFM resolution is set by several factors, including the radius of a tip apex, tip-surface interactions, and environmental conditions.”

Pg. 2 : When discussing the influence of ions on DNA deposition , I would recommend also citing: Pastré et al. (2006). Langmuir 22, 6651–6660.

Corrected. We cited this paper.

Pg. 3 line 148 : The authors write : ‘The major limitation results from the influence of the liquid environment on the cantilever frequency’, do they mean cantilever damping or something else?

Yes, we would like to thank for this comment, we clarified this issue in the text and added an additional reference, please see pg. 4, lines 160-161.

Pg. 4 line 147 : it is unclear what f2 relates to.

Corrected. Please see Figure 1 and its description, pg. 5, line 189-191.

Pg. 6 line 251: I would recommend adding the original images of the DAO-E type lattice, as it it hard to follow the discussion without reading the original paper.

Corrected. We added Figure of the DAO-E type lattice (Figure 3b) and expanded this section to make the text clearer.

Pg. 8 line 307: The authors mention the helical pitch of the DNA 3.51 ± 0.04 nm, without mentioning the expected pitch for the B-form DNA.

Corrected. We added the sentence with appropriate reference: “It was in well agreement with helical pitch of 3.6 nm determined by Rhodes and Klug in enzyme (DNase I) digestion studies [64].”

Pg. 9: Figure 6: panels b & c show DNA –protein complexes at low resolution therefore I would suggest removing them. They are not in line about the discussion of resolving the sub-molecular structure of the DNA.

Corrected. Panels b and c are removed.

Pg. 10 line 340: It is unclear what the bottom and upper panels refer to on figure 8, as there are four of them.

Corrected.

Pg. 10 line 341-44: In this work, AFM was applied for the first time to measure the surface charge density of individual biomolecules, highlighting the capability of AFM in such studies of more complicated systems such as for example DNA-protein complexes.

We simplified this sentence. Please see pg. 11, lines 368-371.

Pg. 12 line 401: The authors mention significant concentration , without giving a rough concentration. Would be nice if it is added.

Corrected as suggested.

Pg 13 line 407: low laser power. Can the authors give an typical range?

Corrected.

Pg 13 line 430: ‘1 per 1010 photons’, I think there is a typeo, since this number is very low.

Corrected. It was a typo. Now it is: 1 per 1010 photons.

Pg 13 line 442: ‘Authors of this work took special attention to prevent …’ it is unclear what is mean by special attention. Can the authors clarify it?

Since it was not clarified in the original work, we removed this part of the sentence.

Pg 14 line 462: When discussing the mode of action of cisplatin, one should also mention that it also interferes with transcription. One should also mention the relevant citation.

Corrected. We added the missing influence of cisplatin on transcription with references.

Pg 14 line 463: The authors mention the magnification of the lens (63x) used in the study. IS it somehow relevant?

The magnification of objective lens was provided only as an additional detail, it is not specifically relevant.

Pg 14 line 474: It is briefly mentioned that silver hydrosol was used as signal amplifier. How does it work? And is it commonly used for Raman measurements?

The silver nanoparticles were used to enhance the Raman scattering cross-section according to the SERS and TERS principle described in page 13-14.

We clarified this issue, please see pg. 15, lines 516-517.

Pg 14 line 491: F missing in ON and OFF.

Corrected.

Pg 15 lines 521-23: It is unclear what plasmid-embedded DNA and plasmid-free DNA mean and what is meant by nucleobase density. The authors should clarify it in the text.

Plasmid-embedded DNA is the circular plasmid molecule having in its sequence of nucleobases an insert of linear DNA fragment of known sequence (β2AR). Plasmid-free DNA is the liner β2AR DNA fragment obtained by enzyme digestion (with HindIII and XbaI) of plasmid molecule.

We added this explanation in the text because it was not clear indeed.

Using the phrase “nucleobase density” we used the exact words of authors of the cited paper. It referred to the number of nucleobases in plasmid molecules that is of course larger than in the short fragment obtained by enzyme digestion. Higher surface density of nucleobases in the case of plasmid-embedded DNA allowed more nucleobases to be probed upon single spectrum acquisition.

We have improved the text adding term “surface density of nucleobases”.

Pg 15 figure 10- X and Y axis on the figure are barely visible. Tip, sample, laser are not indicated on the schematics.

Figure 10 is corrected.

Pg 16 Chapter 6: Is very well written and contains basic description of the techniques, therefore I would recommend placing it before chapter 5. Of course it is optional.

Since this chapter describes the advantages of performing spectroscopic measurements in liquids, we prefer to place it after the chapter describing SERS and TERS working principles.

Pg 16 line 551: ‘’ in liquids that’, should be ‘ in liquid than

Corrected according to suggestion.

Reviewer 2 Report

The manuscript reviews the research on revealing the DNA structure at liquid/solid interfaces using methods of atomic force microscopy imaging and molecular spectroscopy of SERS and TERS. Overall the topic is interesting and the manuscript is well written. It is suggested to be accepted after addressing the following comment.

What is the potential application of these researches on DNA structure at liquid/solid interfaces? It is suggested that authors could elaborate more on this.

Author Response

Re: Submission ID molecules-1418119

Revealing DNA Structure at Liquid/Solid Interfaces by AFM-Based High-Resolution Imaging and Molecular Spectroscopy

Ewelina Lipiec, Kamila Sofińska*, Sara Seweryn, Natalia Wilkosz and Marek Szymonski

Reply to Reviewer 2 comments:

The manuscript reviews the research on revealing the DNA structure at liquid/solid interfaces using methods of atomic force microscopy imaging and molecular spectroscopy of SERS and TERS. Overall the topic is interesting and the manuscript is well written. It is suggested to be accepted after addressing the following comment.

What is the potential application of these researches on DNA structure at liquid/solid interfaces? It is suggested that authors could elaborate more on this.

The section describing applications of research on DNA structure at liquid/solid interfaces is added in chapter ‘7. Concluding remarks and future outlook’:

“Scanning probe microscopy provides extremely high spatial resolution and delivers unique information about DNA topographical features and its properties but the structural information is still limited. Broad perspectives will involve a combination of HR-SPM with other experimental methods, that deliver direct information about chemical structure and composition. One of significant future directions is simultaneous analysis of topographical features and their molecular vibrations via AFM or STM combined with chemically selective molecular spectroscopy such as infrared or Raman. This methodology is universal, and it will be applied in studies of numerous biological systems such as DNA, DNA-proteins or, DNA-drugs complexes, chromatin, multi-component lipid membranes and many others.

This review summarizes advantages of measurements in liquid and physical phenomena responsible for the beneficial role of solvents in molecular nanospectroscopic techniques, highlighting very important future direction in DNA investigations at liquid/solid interface. Measurements in conditions, that mimic nucleic acids natural environment will allow following in the real time biologically significant processes, that are of key importance for life. New findings about the dynamics of structural changes in DNA molecule upon damage, repair and interaction with other biomolecules are fundamentally important in understanding the mysterious role of DNA conformational transitions and chromatin integrity in formation of DNA damage, DNA repair pathways and ability to participate in biochemical reaction crucial for life.”
